# Differential Labeling of Glycoproteins with Alkynyl Fucose Analogs

**DOI:** 10.3390/ijms21176007

**Published:** 2020-08-20

**Authors:** Chenyu Ma, Hideyuki Takeuchi, Huilin Hao, Chizuko Yonekawa, Kazuki Nakajima, Masamichi Nagae, Tetsuya Okajima, Robert S. Haltiwanger, Yasuhiko Kizuka

**Affiliations:** 1Department of Molecular Biochemistry, Nagoya University Graduate School of Medicine, Showa-ku, Nagoya 466-8550, Japan; machenyuyaoxue@yahoo.co.jp (C.M.); htakeuchi@med.nagoya-u.ac.jp (H.T.); tokajima@med.nagoya-u.ac.jp (T.O.); 2Institute for Glyco-Core Research (iGCORE), Nagoya University, Chikusa-ku, Nagoya 464-8601, Japan; 3Complex Carbohydrate Research Center, Department of Biochemistry and Molecular Biology, University of Georgia, Athens, GA 30602, USA; hl.hao@uga.edu (H.H.); rhalti@uga.edu (R.S.H.); 4Center for Highly Advanced Integration of Nano and Life Sciences (G-CHAIN), Gifu University, Gifu 501-1193, Japan; yone1027@gifu-u.ac.jp; 5Center for Research Promotion and Support, Fujita Health University, Toyoake 470-1192, Japan; k-nakaji@fujita-hu.ac.jp; 6Department of Molecular Immunology, Research Institute for Microbial Disease, Osaka University, Suita 565-0871, Japan; mnagae@biken.osaka-u.ac.jp; 7Laboratory of Molecular Immunology, Immunology Frontier Research Center (IFReC), Osaka University, Suita 565-0871, Japan; 8Institute for Glyco-Core Research (iGCORE), Gifu University, Gifu 501-1193, Japan

**Keywords:** fucose, fucosyltransferase, glycan, glycosylation, glycan labeling, click chemistry, notch

## Abstract

Fucosylated glycans critically regulate the physiological functions of proteins and cells. Alterations in levels of fucosylated glycans are associated with various diseases. For detection and functional modulation of fucosylated glycans, chemical biology approaches using fucose (Fuc) analogs are useful. However, little is known about how efficiently each unnatural Fuc analog is utilized by enzymes in the biosynthetic pathway of fucosylated glycans. We show here that three clickable Fuc analogs with similar but distinct structures labeled cellular glycans with different efficiency and protein specificity. For instance, 6-alkynyl (Alk)-Fuc modified *O*-Fuc glycans much more efficiently than 7-Alk-Fuc. The level of GDP-6-Alk-Fuc produced in cells was also higher than that of GDP-7-Alk-Fuc. Comprehensive in vitro fucosyltransferase assays revealed that 7-Alk-Fuc is commonly tolerated by most fucosyltransferases. Surprisingly, both protein *O*-fucosyltransferases (POFUTs) could transfer all Fuc analogs in vitro, likely because POFUT structures have a larger space around their Fuc binding sites. These findings demonstrate that labeling and detection of fucosylated glycans with Fuc analogs depend on multiple cellular steps, including conversion to GDP form, transport into the ER or Golgi, and utilization by each fucosyltransferase, providing insights into design of novel sugar analogs for specific detection of target glycans or inhibition of their functions.

## 1. Introduction

Glycosylation is the most common form of post-translational modifications in mammals, providing vast functional diversity to proteins [1,2]. Various protein functions, such as enzyme activity, proteolysis, subcellular localization, and interaction with other molecules, are greatly influenced by the structures and expression levels of glycans on proteins. The addition or deletion of even a single sugar residue of a glycan in mice results in disease-like phenotypes, including Alzheimer’s disease, cancer, or diabetes [3,4,5]. Furthermore, disease-associated alterations of glycans are clinically used as a biomarker in human disease [6,7]. These findings demonstrate that detection and modulation of specific glycan structures are useful for both basic and clinical research.

Among various glycan structures in mammals, fucosylated glycans regulate various protein functions and are involved in many physiological processes, including development, immunity, and inflammation [8,9]. Mammalian fucosylated glycans are classified into three types: Core fucose (Fuc), terminal Fuc (Lewis (Le) and H antigen), and *O*-Fuc (Figure 1A). Core Fuc is attached to the innermost GlcNAc in *N*-glycans by fucosyltransferase FUT8. Loss of FUT8 in mice results in severe phenotypes, including early death [10], emphysema [10], schizophrenia-like behavior [11], and immune dysfunctions [12,13], demonstrating the physiological importance of core Fuc for various glycoproteins. Moreover, FUT8 is aberrantly upregulated in various types of cancers [14] and identified as a driver for melanoma metastasis [15]. In addition, the removal of core Fuc from therapeutic IgGs dramatically raises their antibody-dependent cellular cytotoxicity (ADCC) activity by ~100 fold [16,17], indicating that modulating core Fuc has important therapeutic value. Le (Le^a^, Le^b^, Le^x^, Le^y^) and H antigens are synthesized by FUT1-7, FUT9, and FUT10 (and possibly FUT11) at *N*- and *O*-glycan terminals. In particular, the functions of sialyl Le^x^ in immunity were extensively studied, as lymphocyte homing is triggered by the interaction between selectin and sialyl Le^x^ [18,19]. Deletion of sialyl Le^x^ or addition of a blocking antibody abolished lymphocyte homing and leukocyte recruitment [20,21]. In addition, the emergence of cancer-specific Le antigens is clinically used as a cancer biomarker, such as sialyl Le^a^ (CA19-9) [22], demonstrating the physiological and pathological significance of Le type Fuc residues. *O*-Fuc glycans are a class of *O*-glycans in which Fuc is directly attached to a Ser/Thr residue within a specific consensus motif of epidermal growth factor-like (EGF) repeats or thrombospondin type 1 repeats (TSRs) (Figure 1A) [8,23,24]. Protein *O*-fucosyltransferase 1 and 2 (POFUT1 and POFUT2) catalyze the *O*-Fuc modification on properly folded EGF repeats and TSRs, respectively, in the endoplasmic reticulum (ER) of cells [25,26]. *O*-Fucosylation regulates multiple biological processes; in particular, *O*-Fuc glycans on EGF repeats located on Notch receptors play a critical role in the Notch-ligand interaction, thus regulating cellular signaling. Elimination of *Pofut1* in mice causes embryonic lethality with otch-related developmental defects [27]. Heterozygous deletion mutations in *POFUT1* cause Dowling-Degos disease in humans [28], and the amplification of *POFUT1* is related to several types of cancers, including hepatocellular carcinoma [29], which strongly suggests the biological significance of *O*-Fuc glycans. Similarly, genetic deletion of *Pofut2* in mice results in embryonic lethality with gastrulation defects mainly due to the loss of secretion of ADAMTS9 matrix protease [30,31]. These studies underscore the biological significance of fucosylated glycans, increasing the need for novel methods to detect and modulate specific fucosylated glycans.

Glycan biosynthesis occurs in the luminal side of the ER and Golgi apparatus and is mediated in a stepwise action by a panel of glycosyltransferases that transfer a sugar residue from a nucleotide sugar (donor) to an appropriate acceptor substrate [32,33,34]. Fucosyltransferases use GDP-Fuc as a donor substrate, which is synthesized by a de novo pathway from GDP-mannose (Man), or by a salvage pathway from free fucose, in the cytosol and transported into ER and Golgi lumen by specific transporters (Figure 1B) [8,9]. The transported GDP-Fuc is subsequently utilized by fucosyltransferases to form fucosylated glycans. Therefore, biosynthesis of fucosylated glycans depends on several factors, including the level and transport of GDP-Fuc as well as efficiency and acceptor specificity of Fuc transfer by fucosyltransferases.

The glycan biosynthetic pathway can also be exploited or shut down by externally supplied sugar analogs, resulting in labeling or functional inhibition of target glycans. In particular, alkynyl- or azide-sugars have been developed and used as glycan probes in combination with click chemistry [35,36,37,38]. Clickable Fuc analogs with an alkyne or azide group were also developed as probes for fucosylated glycans, such as 6-azido-Fuc (6-Az-Fuc), 6-alkynyl-Fuc (6-Alk-Fuc), and 7-alkynyl-Fuc (7-Alk-Fuc) (Figure 1C) [39,40,41,42]. Peracetylated forms of these Fuc analogs are taken into cells, converted to GDP-Fuc analogs by the salvage pathway, transported to the ER or Golgi lumen, and incorporated into proteins by fucosyltransferases (Figure 1B). The labeled glycans having these clickable fucose analogs can be easily detected by chemical ligation [42]. Furthermore, Fuc analogs were also reported to disturb the functions of endogenous fucosylated glycans. For example, 6-Alk-Fuc specifically inhibits a biosynthetic enzyme of GDP-Fuc, FX, resulting in the depletion of endogenous fucosylated glycans [43]. Thio- and fluoro-Fuc were also found to decrease the endogenous levels of fucosylated glycans [44,45,46,47]. More recently, 6-Alk-Fuc and 6-alkenyl-Fuc were demonstrated to be incorporated into *O*-Fuc glycans resulting in inhibition of Notch signaling [48]. These findings indicate that Fuc analogs are useful to detect fucosylated glycans or modulate their functions in cells. However, it is unclear how Fuc analogs are recognized by the biosynthetic enzymes for fucosylated glycans and how efficiently the Fuc analogs are incorporated into different types of fucosylated glycans.

In this study, we focus on three clickable Fuc analogs, 6-Alk-Fuc, 7-Alk-Fuc, and 6-Az-Fuc, and investigate their labeling efficiency for various types of glycoproteins. We also analyzed the conversion of these Fuc analogs to GDP forms in cells and conducted comprehensive in vitro fucosyltransferase assays using nine recombinant enzymes and GDP-Fuc analogs (Figure 1D). Our findings revealed the differences among Fuc analogs in the levels of their GDP forms in cells, utilization by fucosyltransferase, and incorporation into glycans, suggesting that Fuc analogs can be differently and selectively used to label fucosylated glycans or inhibit their functions in cells.

## 2. Results

### 2.1. Differential Incorporation of 6-Alk-Fuc and 7-Alk-Fuc into Glycoproteins

In our previous work, we reported that 7-Alk-Fuc labeled cellular glycans with higher efficiency than 6-Alk-Fuc in mouse embryonic fibroblast (MEF) and HEK293 cells [42]. To examine whether this finding is commonly observed in various cell types, we first treated several cells with peracetylated 6-Alk-Fuc or 7-Alk-Fuc, and the labeled glycans on cellular proteins were detected by click chemistry (Figure 2A). We found that labeling efficiency of these two Fuc analogs was highly cell type-dependent, suggesting that cellular differences in one or more of the enzymes or transporters in the biosynthetic pathway of the fucosylated glycans, or the protein targets, differentially affect incorporation of 6-Alk-Fuc and 7-Alk-Fuc. Furthermore, we found that cellular proteins in HEK293 and A549 cells were preferably labeled with 7-Alk-Fuc, while secreted proteins in the media were labeled with 6-Alk-Fuc with higher efficiency than 7-Alk-Fuc (Figure 2B, left two panels). Secretion of two glycoproteins, laminin alpha 5 and amyloid precursor protein (APP), was unaffected from cells treated with either 6-Alk-Fuc or 7-Alk-Fuc (Figure 2B, right two columns). These data suggest that incorporation of these two Fuc analogs is highly dependent on cell type.

We next compared the incorporation of 6-Alk-Fuc and 7-Alk-Fuc specifically into *O*-Fuc glycans within EGF repeats or TSRs in cells. We expressed and purified NOTCH1 EGF1-18 or thrombospondin 1 (THBS1) TSR1-3 from HEK293T cells incubated with these Fuc analogs. Click chemistry and mass spectral glycoproteomic methods were performed on the purified proteins to examine the incorporation efficiency. 6-Alk-Fuc is known to be utilized by POFUT1/2 and can be incorporated to EGF repeats and TSRs with high efficiency [49]. As expected, we observed the efficient labeling of both NOTCH1 EGF1-18 and THBS1 TSR1-3 with 6-Alk-Fuc (Figure 2C,E). Mass spectral analysis further confirmed the incorporation of 6-Alk-Fuc to *O*-Fuc sites on EGF6 of NOTCH1 EGF1-18 (Figure 2D; Appendix A), and TSR2 (Figure 2F; Appendix A) and TSR3 (Appendix A) of THBS1 TSR1-3. In contrast, 7-Alk-Fuc was not significantly incorporated into either NOTCH1 EGF1-18 or THBS1 TSR1-3 (Figure 2C–F, Appendix A). 6-Alk-Fuc is known as a potent NOTCH1 inhibitor through incorporating into NOTCH1 and interrupting Notch-ligand binding [48]. We performed DLL1-mediated NOTCH1 signaling assays with 6-Alk-Fuc and 7-Alk-Fuc to compare their effects on Notch signaling. 7-Alk-Fuc showed modest NOTCH1 inhibition at high concentrations compared to 6-Alk-Fuc (Appendix A), again suggesting 7-Alk-Fuc was not efficiently incorporated into NOTCH1 EGF repeats. These results show that 6- and 7-Alk-Fuc were differentially incorporated into *O*-fucosylated glycans in cells.

### 2.2. Differences in the Levels of GDP-6-Alk-Fuc and GDP-7-Alk-Fuc in Cells

The above results suggest that differential usage of 6-Alk-Fuc and 7-Alk-Fuc at any step in the biosynthetic pathway of fucosylated glycans could result in the observed differences in incorporation. A candidate step is the conversion of these analogs to GDP forms in cells. We, therefore, treated HEK293 cells with peracetylated 6-Alk-Fuc or 7-Alk-Fuc and quantified the levels of GDP-Fuc analogs and other nucleotide sugars in cells (Figure 3). We found that the level of GDP-6-Alk-Fuc was significantly higher than that of GDP-7-Alk-Fuc, suggesting that GDP-6-Alk-Fuc was more efficiently biosynthesized or more stable than GDP-7-Alk-Fuc in cells, and this difference could explain the weaker labeling of some glycoproteins with 7-Alk-Fuc. A decrease in GDP-Fuc and an increase in its precursor GDP-Man by 6-Alk-Fuc treatment were also found (Figure 3, 2nd graph), which is consistent with our previous report showing that GDP-6-Alk-Fuc selectively inhibits an enzyme for de novo pathway of GDP-Fuc synthesis and depletes endogenous fucosylated glycans [43]. We also observed alterations in the levels of other nucleotide sugars, such as UDP-Xyl and UDP-GlcA, by 6-Alk-Fuc for unknown reasons, suggesting that 6-Alk-Fuc also affects other sugar metabolic pathway in cells.

### 2.3. Labeling Efficiency and Cytotoxicity of 6-Az-Fuc

Another clickable Fuc analog, 6-Az-Fuc, was also examined. Neuro2A cells were treated with peracetylated 6-Az-Fuc as well as 6-Alk-Fuc and 7-Alk-Fuc, and the incorporated Fuc analogs on glycoproteins were labeled and detected (Figure 4A). As a result, 6-Az-Fuc showed highly efficient labeling, compared with 6-Alk- and 7-Alk-Fuc. This suggests that 6-Az-Fuc is highly incorporated into glycans, but the use of two different labeling reagents (biotin-azide for 6-Alk-Fuc and 7-Alk-Fuc, and biotin-alkyne for 6-Az-Fuc) also raised a possibility that this might be caused by different efficiency of click reactions. As we and other groups previously reported, 6-Az-Fuc showed cytotoxicity, which limited availability of this analog in cell and animal experiments [41,42]. We previously examined its cytotoxicity only for Jurkat cells, and in this study, we also evaluated its adverse effects on the growth of adherent cells. We found that peracetylated 6-Az-Fuc showed moderate cytotoxicity toward HEK293 and Neuro2A cells (Figure 4B). As this compound labeled glycans in Neuro2A cells with higher efficiency than the other two analogs, it suggests that despite its cytotoxicity 6-Az-Fuc can also be used as a probe for fucosylated glycans in cells, depending on cell type and glycoproteins.

### 2.4. Tolerance of Fucosyltransferases with Fucose Analogs

The above results clearly showed that three fucose analogs were incoporated into glycans with different efficiency. In mammals, 12 (or 13 if FUT11 is included) fucosyltransferases with different acceptor specificities have been identified to date, and each cell type has different expression profiles of the fucosyltransferases. If these enzymes show different affinity toward GDP-Fuc analogs, that would result in cell type- or glycoprotein-dependent labeling with the Fuc analogs in cells. To explore this possiblity, we measured in vitro activity of 9 purified fucosyltransferases using GDP-Fuc and its analogs (Figure 5 and Appendix A). FUT1, 2, 4, and 9 were expressed as full-length forms in HEK293 cells and immunopurified as described [50], and FUT3 and 5 were similarly expressed in HEK293 cells and purified (Appendix A). FUT8 was expressed as a his-tagged soluble form in COS7 cells and purified from the media [51]. Lacto-N-neotetraose (LNnT) and a biantennary N-glycan (GnGnbiAsn-PNS) were used as an acceptor for FUT1-4 and FUT8, respectively [50,51]. A β4-galactosylated biantennary N-glycan (GGnGGnbi-PA) that was prepared as described previously [50] was used as an acceptor for FUT3 and FUT5 (Appendix A). The activity assay showed that FUT1-5 and FUT8-9 commonly showed the highest activity to GDP-7-Alk-Fuc among the three GDP-Fuc analogs (Figure 5), which is consistent with the previous report [42]. Recombinant his-tagged human POFUT1 (Appendix A) and FLAG-tagged mouse POFUT2 (Appendix A) were expressed in HEK293T cells. Bacterially expressed, C-terminally his-tagged human factor IX-EGF1 and human Thrombospondin 1-TSR3 were used as acceptor substrate for POFUT1 and POFUT2, respectively. Both POFUT1 and POFUT2 utilized three GDP-Fuc analogs at a relatively high efficiency (Figure 5, Appendix A), compared with other fucosyltransferases.

### 2.5. Structural Model of Fucosyltransferases and GDP-Fuc Analogs

Our enzymatic activity assay revealed that FUT1-9 commonly showed lower enzymatic activity toward all Fuc analogs than native GDP-Fuc and selectivity toward GDP-7-Alk-Fuc among the analogs, while POFUT1 and 2 were relatively tolerant of all three analogs (Figure 5). This could be due to structural differences around the donor binding sites of these enzymes. Three crystal structures of human fucosyltransferases complexed with a donor substrate are available at present: FUT8-GDP-acceptor complex, POFUT1-GDP-Fuc complex, and POFUT2-GDP-Fuc complex [52,53,54]. To analyze the local structural differences of these enzymes, we compared the donor binding sites of FUT8, POFUT1, and POFUT2 (Figure 6). The catalytic groove of human FUT8 is relatively shallow, compared with those of POFUT1 and 2. Moreover, the side chain of E373, which is the catalytic base of FUT8 is in close proximity to the methyl group of the Fuc moiety [53,55], suggesting that a donor analog suitable for FUT8 reaction is structurally limited. In contrast, N46, the critical residue of POFUT1 activity [56], and E54, a putative catalytic base of POFUT2, are both located at the opposite side of the catalytic center [52]. Thus, both POFUT1 and 2 have enough space around the methyl group of Fuc, allowing further modification at this position, consistent with the unselective action of these enzymes toward all the Fuc analogs in the activity assay.

## 3. Discussion

We here reported that three clickable Fuc analogs were differentially incorporated into glycoproteins in cells despite their structural similarity. In particular, glycan labeling efficiency of 7-Alk-Fuc was highly variable and dependent on cell type and glycoprotein. Many cellular proteins in HEK293 and A549 cells were efficiently labeled with 7-Alk-Fuc, whereas this analog was poorly incorporated into secreted proteins and *O*-Fuc glycans on EGF and TSR domains. Considering that GDP-7-Alk-Fuc is commonly accepted by all fucosyltransferases tested compared with other analogs, the weak glycan labeling with 7-Alk-Fuc in cells could be derived from the lower ratio of GDP-7-Alk-Fuc to GDP-Fuc in cells than GDP-6-Alk-Fuc to GDP-Fuc.

It is unclear at present why the level of GDP-7-Alk-Fuc in cells was lower than that of GDP-6-Alk-Fuc when the cells were treated with peracetylated 7-Alk-Fuc or 6-Alk-Fuc. Although we cannot rule out the differential rate of incorporation of the compounds through the plasma membrane, this is probably caused by either low efficiency of GDP addition to 7-Alk-Fuc or instability of GDP-7-Alk-Fuc. The addition of GDP to Fuc in the salvage pathway in the cytosol is mediated by sequential actions of fucose kinase and GDP-Fuc pyrophosphorylase [9]. Structural and biochemical analyses of these enzymes with Fuc analogs in the future could elucidate how efficiently GDP-7-Alk-Fuc is produced by the salvage pathway. Regarding the stability, Fuc is considered not to be further catabolized in mammalian cells [9]. However, the metabolism of the unnatural Fuc analogs is poorly understood, raising the possibility that the Fuc analog might be specifically converted to another compound. Isotope labeling of the Fuc analog and its tracing might be required to elucidate how each analog is metabolized in cells. In addition, such metabolites may affect the cellular levels of UDP-Xyl and UDP-GlcA. Indeed, to our surprise, the treatment with peracetylated 6-Alk-Fuc, but not with the treatment with peracetylated 7-Alk-Fuc or a control, specifically decreased the levels of UDP-Xyl and UDP-GlcA in cells. It will also be very interesting to investigate to what extent Xyl- or GlcA-related glycans found on Notch, glycosaminoglycans, and α-dystroglycan are affected by the treatment with Fuc analogs.

A surprising result was that POFUT1 and POFUT2 utilized GDP-7-Alk-Fuc better than or equally well as GDP-6-Alk-Fuc in in vitro assays (Figure 5), but only 6-Alk Fuc was significantly incorporated into EGF repeats and TSRs in cells (Figure 2). Interestingly, 6-Az-Fuc is also not incorporated into EGF repeats by POFUT1 in cells [48] even though POFUT1 utilizes GDP-6-Az-Fuc in vitro (Figure 5). Since POFUT1 and POFUT2 are both localized in the ER [24], a possible reason for this difference could be that GDP-6-Alk-Fuc is more efficiently transported into the ER than either GDP-7-Alk-Fuc or GDP-6-Az-Fuc. GDP-Fuc transport to the Golgi lumen is largely mediated by SLC35C1 [57], but transport into the ER is still puzzling. Although an ER GDP-Fuc transporter has been identified in *Drosophila* [58], the mammalian homolog, SLC35B4, has been shown to be a Golgi-localized, bifunctional transporter for UDP-Xyl and UDP-GlcNAc [59]. Alternatively, knockdown of a different SLC35C1 homolog, SLC35C2, has been reported to affect Notch *O*-fucosylation and function in mammalian cells and is localized in the cis-Golgi and ER-Golgi Intermediate Compartment (ERGIC) [60]. Regardless of which, or both, of these proteins transport GDP-Fuc into the ER, the transport efficiency of GDP-7-Alk-Fuc or GDP-6-Az-Fuc to the ER lumen might be lower than GDP-6-Alk-Fuc. Although it is extremely difficult to measure the levels of nucleotide sugars in each organelle at present, future in-depth biochemical and structural studies could clarify the specificity of these GDP-Fuc transporters toward each analog.

Our structural modeling demonstrated that POFUTs have larger space around the Fuc binding sites, suggesting that these enzymes can accommodate various Fuc analogs having a chemical group at the C6-position. Although the atomic structures of human FUT1-7 and 9 are not yet available, the fact that they all use GDP-7-Alk-Fuc more efficiently than GDP-6-Alk-Fuc suggest that the basic architecture around donor binding sites is similar to FUT8. Future structural studies would unravel why these enzymes are tolerant of only limited Fuc analogs.

As fucosylated glycans have distinct functions and are involved in specific diseases, labeling or inhibition of specific types of fucosylated glycans would be highly desirable. Although it could also be possible that Fuc analogs uniformly label or inhibit all fucosylated glycans, our present data clearly show that each Fuc analog is differently used in cells, dependent on cell type and glycoprotein. In particular, the fact that POFUTs can structurally and biochemically tolerate various analogs compared with other FUTs suggest that new POFUTs-selective Fuc analogs can be developed. For this purpose, chemical synthesis of new Fuc analogs and their biochemical investigation and metabolic tracing in combination with structural biology will be needed.

## 4. Materials and Methods

### 4.1. Materials

Peracetylated 6-Alk-Fuc and 7-Alk-Fuc were purchased from Peptide Institute Inc. (Ibaraki, Japan), and peracetylated 6-Az-Fuc was purchased from Synthose Inc. (Concord, ON, Canada). GDP-Fuc analogs were synthesized as described previously [42]. GDP-Fuc was purchased from Sigma (St. Louis, MO, USA). Anti-laminin alpha 5 Ab was from Abcam (Cambridge, UK) (ab77175), and anti-APP mAb clone 22C11 (MAB348) and anti-GAPDH Ab (MAB374) were from Millipore (Darmstadt, Germany).

### 4.2. Cell Culture

Caco2, HEK293, A549, and Neuro2A cells were cultured in DMEM supplemented with 10% FBS and kanamycin. Molt4 and Namalwa were cultured in RPMI supplemented with 10% FBS. All cells were obtained from ATCC (Manassas, VA, USA). 

### 4.3. Plasmid Construction

cDNAs encoding human FUT3 and 5 were amplified by PCR with the primers below (FUTs-For and -Rev) using cDNA libraries of Hep3B and HEK293 cells as a template, respectively. The amplified fragments were inserted into pCR-Blunt using Zero Blunt PCR Cloning Kit (ThermoFischer Scientific). cDNAs for FUT3 and 5 skipping their stop codons were amplified by PCR with the common primers below (pcDNA-For and -Rev), digested with EcoRI and XhoI, and ligated into the EcoRI/XhoI site of pcDNA6/myc-His A. FUT3-For, GATACTCTGACCCATGGATCCCCTG; FUT3-Rev, AGGTCCCAGGTAAGAGAGAGGGTTG; FUT5-For, TAGCTACTCTGACCCATGGATCCC; FUT5-Rev, TAGGTAGGTGAGGCCCTGGGAAAGT; pcDNA-For, AAAGAATTCTGACCCATGGATCCCCTGGG; pcDNA-Rev, TTTCTCGAGGGTGAACCAAGCCGCTATGC. The expression plasmids for mouse FUT1, 2, 4, and 9 were constructed as described previously [42], and the plasmid for the soluble form of human FUT8 was constructed as described previously [43].

### 4.4. Labeling and Detection of Fucosylated Glycans in Cells and Media

#### 4.4.1. Labeling

Cells were treated with DMSO or 100 μM peracetylated 6-Alk-, 7-Alk-, or 6-Az-Fuc for 24 h, and the labeled proteins in cells and media were biotinylated using Click-iT Protein Reaction Buffer Kit (ThermoFischer Scientific, Waltham, MA, USA) and Biotin-alkyne (ThermoFischer Scientific) or Biotin-azide (ThermoFischer Scientific) as described previously [42,43]. Proteins secreted into culture media were precipitated with EtOH as follows. After removing cell debris by centrifugation at 1000× *g* for 5 min, 1/30 volume of 5 M NaCl and 1/2.5 volume of ice-cold EtOH were added to the media, followed by incubation at −30 °C for 20 min. Precipitated proteins were collected by centrifugation at 15,000 × *g* for 20 min, washed with 70% EtOH, and centrifuged at 15,000× *g* for 5 min. The precipitated proteins were lysed and biotinylated, as in the case for cellular proteins.

#### 4.4.2. SDS-PAGE, Detection of Biotinylated Proteins, and Western Blot

The biotinylated proteins were separated by 5–20% SDS-PAGE, transferred to a nitrocellulose membrane, and detected using VECTASTAIN ABC Standard Kit (Vector Laboratories, Burlingame, CA, USA). For Western blotting, cells were lysed with TBS containing 0.5% nonidet P-40 and a protease inhibitor cocktail and sonicated. Proteins secreted into culture media were concentrated by Amicon Ultra-4 10 kDa (Millipore) after removing cell debris by centrifugation at 1000× *g* for 5 min. Protein concentrations in the cell lysates and media were measured using a Pierce BCA Protein Assay Kit (ThermoFisher Scientific), and the proteins were denatured in a Laemmli sample buffer. Proteins were separated by 5–20% SDS-PAGE, transferred to nitrocellulose membranes. After blocking with TBS containing 5% skim milk and 0.05% Tween-20, the membranes were incubated with 1st Ab and subsequently with HRP-labeled 2nd Ab. Signals were visualized with Western Lightning Plus-ECL (PerkinElmer, Waltham, MA, USA) or SuperSignal West Femto Maximum Sensitivity substrate (Thermo Fisher Scientific) using FUSION SOLO 7s EDGE (Vilber-Lourmat, Eberhardzell, Germany).

### 4.5. Labeling and Detection of O-Fuc Glycans

#### 4.5.1. Production of 6- or 7-Alk-Fuc-Modified mNOTCH1 EGF1-18 and hTHBS1 TSR1-3

HEK293T cells were seeded in 10 cm dishes and grown to 70% confluency in Dulbecco’s modified eagle medium (DMEM) (ThermoFischer Scientific) with 10% FBS and 1% penicillin/streptomycin. Cells were transiently transfected with 2 µg/dish pSecTag2c-mNOTCH1-EGF1-18-Myc-His_6_ [61] or pSecTag2c-hTHBS1-TSR1-3-Myc-His_6_ [49] with 12 µL/dish polyethyleneimine (PEI). After 6 h, the medium was changed to an 8 mL/dish Opti-MEM (GIBICO, Life Technologies) with 50 µM peracetylated 6- or 7-Alk-Fuc or an equal volume of DMSO. After 3 (for hTHBS1 TSR1-3) or 4 days (for mNOTCH1 EGF1-18), the medium was collected. mNOTCH1 EGF1-18 and hTHBS1 TSR1-3 proteins were purified using Ni-NTA resin (Qiagen, Hilden, Germany).

#### 4.5.2. Biotinylation and Detection of 6- or 7-Alk-Fuc-Modified mNOTCH1 EGF1-18 and hTHBS1 TSR1-3 Proteins

Cu(I)-catalyzed azide-alkyne cycloaddition (CuAAC) was performed on purified mNOTCH1 EGF1-18 and hTHBS1 TSR1-3 proteins (~100 ng) to biotinylate proteins modified with 6- or 7-Alk-Fuc. The reaction was performed with 100 µM Biotin picolyl azide (Click Chemistry Tools, Scottsdale, AZ, USA), 500 µM BTTP (3-(4-((bis((1-tertbutyl)-1H-1,2,3-triazol-4-yl)methyl)amino)methyl)-1H-1,2,3-triazol-1-yl)-propan-1-ol, a gift from Dr. Peng Wu) [62], 250 µM copper sulfate, 2.5 mM sodium ascorbate in water and gently shaking at room temperature for one hour. The biotinylated samples were then analyzed by Western blot. Samples were loaded onto 4–15% SDS-PAGE, transferred to nitrocellulose membrane and probed with anti-Myc antibody (Clone 9E10, Invitrogen, 1:2000), followed by IRDye 680-conjugated goat anti-mouse IgG antibody (LI-COR, 1:5000) to detect the total protein amount. To detect the biotinylated proteins, the nitrocellulose membrane was probed with IRDye 680RD streptavidin (LI-COR, 1:2000) or IRDye 800CW streptavidin (LI-COR, 1:2000). The Western blot bands were visualized using an Odyssey System (LI-COR).

#### 4.5.3. Mass Spectrometry of 6- or 7-Alk-Fuc-Modified mNOTCH1 EGF1-18 and hTHBS1 TSR1-3 Proteins

Purified proteins (~3 µg) from each condition were reduced and alkylated as described [63] and loaded onto 4–15% SDS-PAGE. The gels were stained with GelCode Blue (Thermo Scientific). Desired bands were cut and washed overnight with gel wash solution (50% methanol in 20 mM diammonium phosphate, pH 8.0) to remove SDS. The proteins were digested (in-gel) with V8 (for mNOTCH1 EGF1-18) or double-digested with trypsin and chemotrypsin (for hTHBS1 TSR1-3) as described [63]. The resulting peptides were analyzed by LC-MS/MS using an EasyNano-LC with a C18 EasySpray PepMap RSLC C18 column (50 μm × 15 cm, Thermo Fisher Scientific) coupled to a Thermo Fischer Q-Exactive Plus mass spectrometer. The data processing was performed using Proteome Discoverer 2.1 software (Thermo Scientific) with Byonic (Protein Metrics, Cupertino, CA, USA). For each peptide glycoform, the ion with the highest Byonic score was chosen for the generation of extracted ion chromatograms (EICs). The EICs were smoothed using a Gauss algorithm. EICs show the relative amount of each glycoform of a given peptide. The ion intensity for each glycoform was extracted from the MS1 spectrum by their respective *m/z* and plotted against retention time. When the EICs of different glycoforms from a given peptide are overlaid, by comparing their relative peak intensities, we can know the relative abundance of each glycoform of the analyzed peptide.

### 4.6. Nucleotide Sugar Analysis

Nucleotide sugars from HEK293 cells cultured in the presence of peracetylated 6-Alk-Fuc or 7-Alk-Fuc (50 μM) or DMSO for 24 h were prepared and quantified by hydrophilic interaction liquid chromatography-electrospray ionization-tandem mass spectrometry (HILIC-ESI-MS/MS) as reported previously [64,65]. Briefly, cells plated on a 10 cm dish were washed once in cold PBS and collected in ice-cold 70% ethanol (2 mL). The extract was centrifuged at 16,000× *g* for 15 min at 4 °C, and the supernatant was lyophilized. The freeze-dried sample was dissolved in 1 mL of 10 mM NH_4_HCO_3_ and applied to an Envi-Carb column [65].

LC-ESI-MS/MS was performed on an LCMS-8060 (Shimadzu, Kyoto, Japan) coupled with Nexcera HPLC system (Shimadzu, Kyoto, Japan). Chromatography was performed on ZIC column with phosphocholine phase (ZIC-cHILIC, 2.1 mm i.d. × 150 mm, 3 µm; Merck SeQuant, Umea, Sweden) [64]. The nucleotide sugars were quantified based on the peak area of the major fragments in the multiple reaction monitoring (MRM) modes using specific precursor-product ion pairs as follows: *m/z* 613(Q1)→322 (Q3) for CMP-NeuAc, *m/z* 565→323 for UDP-Gal and UDP-Glc, *m/z* 606→385 for UDP-HexNAc, *m/z* 604→442 for GDP-Man, *m/z* 579→323 for UDP-GlcA, *m/z* 588→442 for GDP-Fuc, *m/z* 599→442 for GDP-6-Alk-Fuc [43], and *m/z* 612→442 for GDP-7-Alk-Fuc [42]. Standard nucleotide sugars with known concentrations were used for quantification. The nucleotide sugar levels were normalized as pmol/mg protein.

### 4.7. Cell Proliferation Assay

Cell growth was examined using Cell Counting Kit-8 (Dojindo, Kumamoto, Japan), according to the manufacturer’s protocol. In brief, 5000 cells were seeded on a well of 96-well plate in the presence or absence of 100 μM peracetylated Fuc analogs, followed by incubation at 37 °C in a CO_2_ incubator. At each time point, 10 μL of CCK-8 reagent (supplied in the kit) was added to a well, followed by 1 h incubation at 37 °C in a CO_2_ incubator and measurement of absorbance at 450 nm.

### 4.8. Activity Assay of FUTs

Expression and purification of soluble His-tagged human FUT8, myc-tagged full-length mouse FUT1, 2, 4, and 9 were conducted as described previously [42,50]. The activity of FUT1, 2, 4 toward pyridylamine (PA)-labeled LNnT and that of FUT8 toward PA-labeled GlcNAc-terminated biantennary *N*-glycan (GnGnbiAsn-PNS) were measured as described previously [42]. Myc-tagged full length human FUT3 and 5 were expressed in HEK293 cells and immunopurified from the cell lysates with anti-Myc 4A6 Ab as reported previously [42]. The activity of FUT3 and 5 bound to the beads toward PA-labeled galactose-terminated biantennary *N*-glycan (GGnGGnbi-PA) was measured as described previously [50]. Substrates and products were analyzed by reversed-phase HPLC equipped with a TSK-gel ODS-80TM (4.6 × 150) column (TOSOH). Isocratic mobile phase (A, 20 mM ammonium acetate pH 4.0; B, 20 mM ammonium acetate pH 4.0 containing 1% 1-butanol) was used, and the appropriate buffer B contents were adjusted for each enzyme assay. For each assay, 100 pmol of PA-LNnT or GnGnbiAsn-PNS or 25 pmol of GGnGGnbi-PA was used, and the reaction mixtures were incubated at 37 °C for 3 h (FUT1-5 and 9) or 30 min (FUT8). The amount of transferred Fuc was quantified by comparing the peak area of the product in HPLC.

### 4.9. Activity Assay of POFUTs

#### 4.9.1. Preparation and Purification of EGF Repeats and THBS1-TSR3 Repeats

The pET20b(+)−based human factor IX EGF1 expression plasmid was described [66]. The pET20b(+)−based human THBS1-TSR3 expression plasmid was described [67]. Recombinant EGF repeat and TSR proteins were expressed in *Escherichia coli* BL21(DE3) cells and purified by Ni-NTA agarose (Wako) and RP-HPLC as previously described with a slight modification [68].

#### 4.9.2. Expression and Purification of Recombinant POFUT1 and POFUT2 Proteins

The pSecTag2c-based N-terminally 6xHis-tagged human POFUT1 lacking C-terminal RDEF sequence expression plasmid was described previously [69]. The pCMV6-based Myc-DDK-tagged mouse protein *O*-fructosyltransferase 2 (POFUT2) expression plasmid was purchased from Origene.

Recombinant POFUT1 and POFUT2 proteins were overexpressed in HEK293T cells by transient transfection of the expression plasmids using PEI as previously described [70]. As for POFUT1, the culture medium from the transfected cells was collected and purified by Ni-NTA agarose. Briefly, the culture media was applied to Ni-NTA agarose with gravity flow. After washing twice with 10 mL of Tris-HCl pH 7.4 containing 0.65 M NaCl and 10 mM imidazole, then proteins were eluted with 1 mL of TBS containing 250 mM imidazole and dialyzed against TBS containing 20% glycerol at 4 °C overnight. The protein concentration of POFUT1 was determined by 10% SDS-PAGE followed by Coomassie staining using BSA as a standard. As for mouse POFUT2, the transiently transfected cells were harvested by TBS pH 7.4 containing 1% Triton X-100 and protease inhibitor cocktail and incubated on ice for 60 min. After centrifugation at 14,800 rpm at 4 °C for 10 min, the supernatant (1 mL) was collected. The cell lysates were mixed with 40 μL pre-equilibrated Anti-DDDDK (FLAG) gel and rotated at 4 °C overnight. The gel was washed with 200 μL of the wash buffer, which consisted of 20 mM Tris-HCl pH7.5 containing 300 mM NaCl and 0.1% Nonidet P-40 for 3 times. The bound POFUT2 proteins were eluted with 40 μL of 0.3 mg/mL 3× DDDDK-tag peptides twice. To remove DDDDK-tag peptides, the samples were centrifuged with Amicon Ultra 0.5 mL Centrifugal Filters 30K. Following 10% SDS-PAGE, the Western-blotting analysis was performed using an anti-FLAG primary antibody (MBL) to confirm the presence of FLAG-tagged POFUT2. Coomassie staining using BSA as a standard was also performed to determine the purity and concentration of POFUT2.

#### 4.9.3. Determination of POFUTs’ Activities towards GDP-Fuc Analogs

To determine the activity of different Fuc analogs as a donor substrate, an in vitro POFUTs reaction was performed, the reaction products were analyzed by RP-HPLC. The reaction mixture (30 μL) contained 50 mM HEPES pH 7.0, 10 mM MnCl_2_, 0.5% Nonidet P-40, 10 μM GDP-Fuc or GDP-6-Alk-Fuc, GDP-7-Alk-Fuc, GDP-6-Az-Fuc as donor substrates, 10 μM acceptor substrates and purified POFUTs. For the reactions with POFUT1, EGF1 proteins from human factor XI were used as acceptor substrates. For the reactions with POFUT2, TSR3 proteins from human THBS1 was used as acceptor substrates. The reaction was incubated at 37 °C for 30 min and stopped by addition of 970 μL Milli-Q water. The reaction mixtures were analyzed by reverse-phase HPLC on a 250 × 4.6 mm InertSustainSwift C18 column using a 30 min linear (60 min for POFUT2 reaction products) gradient from 20–38% of Solvent B (0.1% trifluoroacetic acid in acetonitrile) in solvent A (0.1% trifluoroacetic acid in water), and the absorbance was monitored at 210 nm. The intensity of peaks was used for quantification (Appendix A). The fractions containing modified or unmodified proteins were collected, vacuum-dried, and subjected to the LC-MS analysis for the confirmation of the addition of Fuc analogs. The LC-MS analysis was performed as previously described [70].

### 4.10. Structural Modeling of GDP-Fucose Analogs and Fucosyltransferases

All 3D structural figures were drawn with program PyMOL ver 2.0 (The PyMOL Molecular Graphics System, Version 2.0, Schrödinger, LLC, New York, NY, USA). Structural superposition was performed with program SUPERPOSE [71].

## Figures and Tables

**Figure 1 ijms-21-06007-f001:**
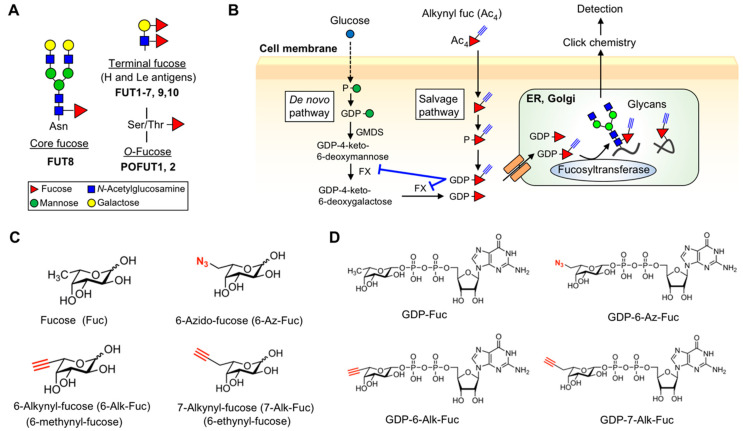
Structure and biosynthesis of fucosylated glycans. (**A**) Structures of fucosylated glycans in mammals. Note that Lewis (Le)-type glycans can be fucosylated on the Gal or GlcNAc, or both. (**B**) Schematic drawing of the biosynthetic pathway of fucosylated glycans in cells. GDP-6-Alk-Fuc inhibits FX. Ac_4_ indicates a peracetylated form. (**C**) Chemical structures of Fuc and its analogs used in this study. (**D**) Chemical structures of GDP-Fuc and its analogs used in this study.

**Figure 2 ijms-21-06007-f002:**
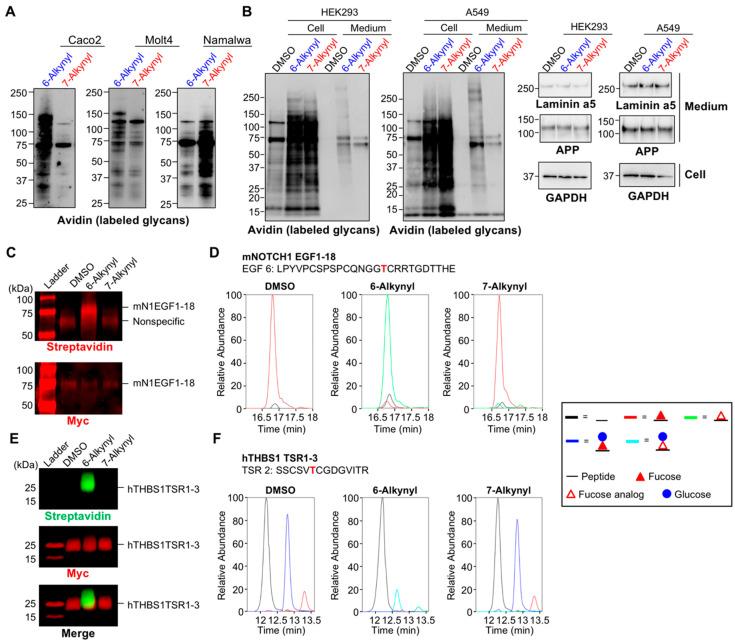
Differential glycoprotein labeling with 6-Alk-Fuc and 7-Alk-Fuc. (**A**) Caco-2, Molt4, and Namalwa cells were treated with peracetylated 6-Alk-Fuc, 7-Alk-Fuc, or DMSO. Incorporated Fuc analogs in the cell lysates were biotinylated by click chemistry, and the labeled glycans on proteins were detected by blotting with HRP-streptavidin. (**B**) HEK293 and A549 cells were treated with peracetylated 6-Alk-Fuc, 7-Alk-Fuc, or DMSO. Left 2 panels: Incorporated Fuc analogs in the cell lysates or secreted proteins were biotinylated by click chemistry, and the labeled glycans on proteins were detected by blotting with HRP-streptavidin. Right 2 columns: Proteins in the cell lysates and secreted into the culture media were western blotted with anti-laminin alpha 5, anti-APP, or anti-GAPDH. (**C**) HEK293T cells were transfected with mouse NOTCH 1 (mN1) EGF1-18 plasmid and incubated with 50 μM peracetylated 6-Alk-Fuc or 7-Alk-Fuc for 4 days. mNOTCH1 EGF1-18 was purified from media and subjected to CuAAC with azido-biotin probe to examine Fuc analog incorporation. The samples were analyzed by Western blot. Top panel: Probed with streptavidin; bottom panel, probed with anti-Myc. (**D**) mN1 EGF1-18 was prepared as in C, digested with V8 protease, and the resulting peptides analyzed by mass spectrometry. An extracted ion chromatogram (EIC) of glycoforms of a peptide from mNOTCH1 EGF6 was prepared. Spectra for these ions are in Appendix A. Black line, unmodified; red line, Fuc modified; green line, 6-Alk-Fuc or 7-Alk-Fuc modified. (**E**) HEK293T cells were transfected with hTHBS1 TSR1-3 plasmid and incubated with 50 μM peracetylated 6-Alk-Fuc or 7-Alk-Fuc for 3 days. hTHBS1 TSR1-3 was purified from secreted media and subjected to CuAAC with azido-biotin probe to examine Fuc analog incorporation. The samples were analyzed by Western blot. Top panel: Probed with streptavidin; middle panel, probed with anti-Myc; bottom panel, merged. (**F**) hTHBS1 TSR1-3 was prepared as in E, digested with trypsin and chymotrypsin, and the resulting peptides analyzed by mass spectrometry. An EIC of the different glycoforms of a peptide from hTHBS1 TSR2 was prepared. Spectra for these ions are in Appendix A. Black line, unmodified; red line, Fuc modified; blue line, glucose (Glc)-Fuc modified; green line, 6-Alk-Fuc or 7-Alk-Fuc modified; aqua line, Glc-6-Alk-Fuc or Glc-7-Alk-Fuc.

**Figure 3 ijms-21-06007-f003:**
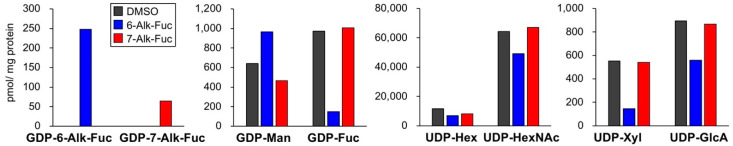
HEK293 cells were treated with 100 μM peracetylated 6-Alk-Fuc, 7-Alk-Fuc, or DMSO for 24 h. The levels of GDP-Fuc analogs and other nucleotide sugars were quantified (*n* = 2).

**Figure 4 ijms-21-06007-f004:**
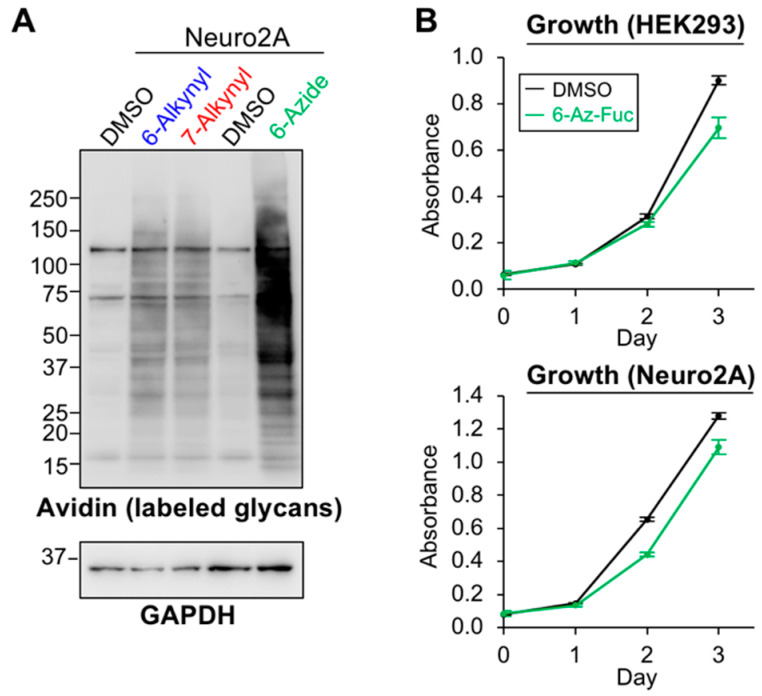
Labeling efficiency and cytotoxicity of 6-Az-Fuc. (**A**) Neuro2A cells were treated with DMSO, 100 μM peracetylated 6-Alk-Fuc, 7-Alk-Fuc or 6-Az-Fuc. Incorporated Fuc analogs in the cell lysates were biotinylated by click chemistry, and the labeled glycans on proteins were detected by blotting with HRP-streptavidin. For labeling, the same concentrations of biotin-azide (lane 1–3) and biotin-alkyne (lane 4 and 5) were used. (**B**) Growth of HEK293 and Neuro2A cells was examined after addition of DMSO or peracetylated 6-Az-Fuc at Day 0 (*n* = 3). All graphs show means ± SD.

**Figure 5 ijms-21-06007-f005:**
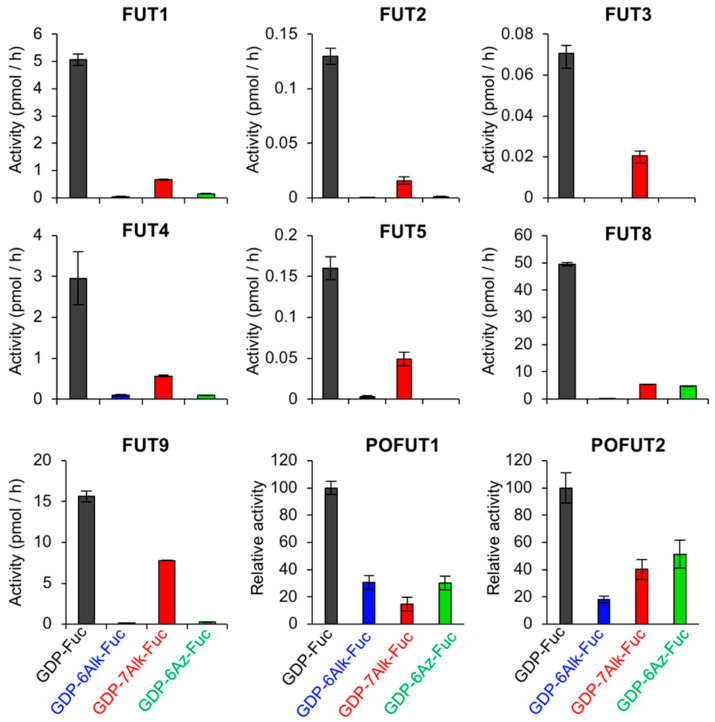
In vitro fucosyltransferase activity using GDP-Fuc and its analogs. Nine recombinant fucosyltransferases (FUT1, 2, 3, 4, 5, 8, 9, and POFUT1, and 2) were expressed and purified from mammalian cells. The activity of FUT1, 2, 4, 9 toward LNnT (*n* = 3), that of FUT3 and 5 toward GGnGGnbi-PA (*n* = 3), that of FUT8 toward GnGnbiAsn-PNS (*n* = 3), that of POFUT1 toward EGF1 from human factor IX, and that of POFUT2 toward TSR3 from human Thrombospondin 1 were measured. All graphs show means ± SD.

**Figure 6 ijms-21-06007-f006:**
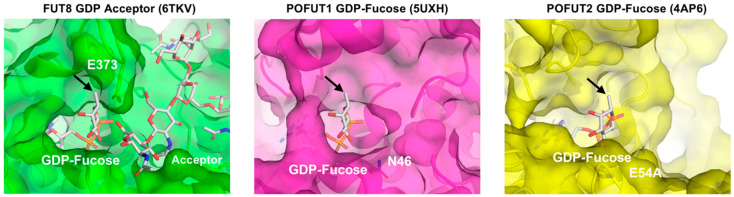
Structural comparison among FUT8-GDP-acceptor complex (PDB code: 6TKV, green), POFUT-GDP-Fuc complex (PDB code: 5UXH, magenta) and POFUT2-GDP-Fuc complex (PDB code: 4AP6, yellow). Protein molecules are shown in semi-transparent surface and ribbon models. Acceptor and GDP-Fuc are shown in stick models. The catalytic residues, E373 (FUT8), N46 (POFUT1) and E54 (POFUT2), located at the rim of catalytic center are labeled. An arrow indicates C6 of Fuc.

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
