# Peer review of "Differential Labeling of Glycoproteins with Alkynyl Fucose Analogs"

_ijms, 2020, doi:10.3390/ijms21176007_

Round 1

Reviewer 1 Report

The present manuscript by Prof. Kizuka deals with the study of three different types of clickable fucose derivatives. The introduction section is quite remarkable and makes clear the soundness of these studies. Furthermore, the research reported here serves to point out the existing differences in labeling properties of the three analogs (two different alkynes and one azido fucosyl derivative) with respect to different cell types. In my opinion, the reported work is very relevant, has been carried out correctly, and deserves to be published in IJMS.

Author Response

We really appreciate your positive comments.

Reviewer 2 Report

I think this is a very useful and important study for the use of metabolic labelling of glycans/glycoproteins. The authors show very convincingly that there is differential labelling of the 6- and 7-alkynylfucose sugars across a variety of proteins with different glycosylation. This is significant because it brings into question global quanitfication of glycans/glycoproteins using these methods. Clearly careful controls are required to evaluate differential labelling. The mechanistic studies looking at different levels of GDP-Fuc analogues and their use as substrates of glycosyltransferases is also quite insightful. Clear trends to suggest that the 6-alkynyl Fuc is more tolerated by the enzymes amking GDP-Fuc, whereas the converse may be true with some FucT enzymes. I also liked the way the paper was written - very clear and in depth introduction. A couple of minor points that could be improved:

It is not entirely clear to me how the quantification of the GDP-Fuc mass spec was doen - the fragmentation experiement is clearly for qualitative identification of the nucleotide sugars, but how was quantification done? on the parent ions with standards of known concentration?

The 6-azido sugar clearly labels more effectively, but is this a result of degree of incorporation or is it because the click reaction is using a different reagent - alkynyl biotin rather than azidobiotin? Some clarification here would be appreciated.

Author Response

Comment 1

I think this is a very useful and important study for the use of metabolic labelling of glycans/glycoproteins. The authors show very convincingly that there is differential labelling of the 6- and 7-alkynylfucose sugars across a variety of proteins with different glycosylation. This is significant because it brings into question global quanitfication of glycans/glycoproteins using these methods. Clearly careful controls are required to evaluate differential labelling. The mechanistic studies looking at different levels of GDP-Fuc analogues and their use as substrates of glycosyltransferases is also quite insightful. Clear trends to suggest that the 6-alkynyl Fuc is more tolerated by the enzymes amking GDP-Fuc, whereas the converse may be true with some FucT enzymes. I also liked the way the paper was written - very clear and in depth introduction. A couple of minor points that could be improved:

Response

Thank you very much for your positive comments. We have revised the paper according to your questions. We hope the revised paper is now acceptable.

Comment 2

It is not entirely clear to me how the quantification of the GDP-Fuc mass spec was doen - the fragmentation experiement is clearly for qualitative identification of the nucleotide sugars, but how was quantification done? on the parent ions with standards of known concentration?

Response

Thank you for pointing it out. To more clearly show how we quantified GDP-Fuc and related molecules, we have revised the paper as shown below.

For the analysis of nucleotide sugars, we used standard nucleotide sugars with known concentrations, and compared the peak area of the major fragment ions (for example [GDP-H]-for GDP-Fuc). We have modified the sentence “The nucleotide sugars were quantified based on the peak area of the major fragments” (line 445) and added a sentence, “Standard nucleotide sugars with known concentrations were used for quantification.” (line 450) in section 4.6. For MS analysis of glycopeptide modified with O-Fuc, we have added a couple of sentences as follows, “EICs show the relative amount of each glycoform of a given peptide. The ion intensity for each glycoform was extracted from MS1 spectrum by their respective m/z and plotted against retention time. When the EICs of different glycoforms from a given peptide are overlaid, by comparing their relative peak intensities, we can know the relative abundance of each glycoform of the analyzed peptide.” (line 428-432) in 4.5.3 section. For FUT activity assay, we used HPLC and quantified the transferred Fuc by looking at the peak area of the products. We have added a sentence, “The amount of transferred Fuc was quantified by comparing the peak area of the product in HPLC.” (line 473-474) in 4.8 section.

Comment 3

The 6-azido sugar clearly labels more effectively, but is this a result of degree of incorporation or is it because the click reaction is using a different reagent - alkynyl biotin rather than azidobiotin? Some clarification here would be appreciated.

Response

Thank you very much. We totally agree that the higher signals were caused by those two possibilities. We took two DMSO controls in Fig. 4A, and biotin-azide and biotin-alkyne were used for DMSO control in lane 1 and 4, respectively. At least we confirmed that the non-specific signals derived from these control click reactions were almost the same between these two click reagents. We have added a sentence in the legend, “For labeling, the same concentrations of biotin-azide (lane 1-3) and biotin-alkyne (lane 4 and 5) were used.” We also added a couple of sentences in the Results, “This suggests that 6-Az-Fuc is highly incorporated into glycans, but the use of two different labeling reagents (biotin-azide for 6-Alk-Fuc and 7-Alk-Fuc, and biotin-alkyne for 6-Az-Fuc) also raised a possibility that this might be caused by different efficiency of click reactions.” (line 208-211)